# Lifted Symmetry Detection and Breaking for MAP Inference

**Tim Kopp**
University of Rochester
Rochester, NY
tkopp@cs.rochester.edu

**Parag Singla**
I.I.T. Delhi
Hauz Khas, New Delhi
parags@cse.iitd.ac.in

**Henry Kautz**
University of Rochester
Rochester, NY
kautz@cs.rochester.edu

## Abstract

Symmetry breaking is a technique for speeding up propositional satisfiability testing by adding constraints to the theory that restrict the search space while preserving satisfiability. In this work, we extend symmetry breaking to the problem of model finding in weighted and unweighted relational theories, a class of problems that includes MAP inference in Markov Logic and similar statistical-relational languages. We introduce term symmetries, which are induced by an evidence set and extend to symmetries over a relational theory. We provide the important special case of term equivalent symmetries, showing that such symmetries can be found in low-degree polynomial time. We show how to break an exponential number of these symmetries with added constraints whose number is linear in the size of the domain. We demonstrate the effectiveness of these techniques through experiments in two relational domains. We also discuss the connections between relational symmetry breaking and work on lifted inference in statistical-relational reasoning.

## 1 Introduction

Symmetry-breaking is an approach to speeding up satisfiability testing by adding constraints, called *symmetry-breaking predicates* (SBPs), to a theory [7, 1, 16]. Symmetries in the theory define a partitioning over the space of truth assignments, where the assignments in a partition either all satisfy or all fail to satisfy the theory. The added SBPs rule out some but not all of the truth assignments in the partitions, thus reducing the size of the search space while preserving satisfiability.

We extend the notion of symmetry-breaking to model-finding in relational theories. A relational theory is specified by a set of first-order axioms over finite domains, optional weights on the axioms or predicates of the theory, and a set of ground literals representing evidence. By model finding we mean satisfiability testing (unweighted theories), weighted MaxSAT (weights on axioms), or maximum weighted model finding (weights on predicates). The weighted versions of model finding encompass MAP inference in Markov Logic and similar statistical-relational languages.

We introduce methods for finding symmetries in a relational theory that do not depend upon solving graph isomorphism over its full propositional grounding. We show how graph isomorphism can be applied to just the evidence portion of a relational theory in order to find the set of what we call *term symmetries*. We go on to define the important subclass of *term equivalent symmetries*, and show that they can be found in $O(nM \log M)$ time where $n$ is the number of constants and $M$ is the size of the evidence.

Next we provide the formulation for breaking term and term equivalent symmetries. An inherent problem in symmetry-breaking is that a propositional theory may have an exponential number of symmetries, so breaking them individually would increase the size of the theory exponentially. This is typically handled by breaking only a portion of the symmetries. We show that term equivalent symmetries provide a compact representation of exponentially many symmetries, and an exponen-

tially large subset of these can be broken by a small (linear) number of SBPs. We demonstrate these ideas on two relational domains and compare our approach to other methods for MAP inference in Markov Logic.

## 2   Background

**Symmetry Breaking for SAT**   Symmetry-breaking for satisfiability testing, introduced by Crawford et. al.[7], is based upon concepts from group theory. A *permutation* $\theta$ is a mapping from a set $L$ to itself. A permutation group is a set of permutations that is closed under composition and contains the identity and a unique inverse for every element. A literal is an atom or its negation. A clause is a disjunction over literals. A CNF theory $\mathcal{T}$ is a set (conjunction) of clauses. Let $L$ be the set of literals of $\mathcal{T}$. We consider only permutations that respect negation, that is $\theta(\neg l) = \neg\theta(l)$ ($l \in L$). The *action* of a permutation on a theory, written $\theta(\mathcal{T})$, is the CNF formula created by applying $\theta$ to each literal in $\mathcal{T}$. We say $\theta$ is a *symmetry* of $\mathcal{T}$ if it results in the same theory *i.e.* $\theta(\mathcal{T}) = \mathcal{T}$.

A model $M$ is a truth assignment to the atoms of a theory. The action of $\theta$ on $M$, written $\theta(M)$, is the model where $\theta(M)(P) = M(\theta(P))$. The key property of $\theta$ being a symmetry of $\mathcal{T}$ is that $M \models \mathcal{T}$ iff $\theta(M) \models \mathcal{T}$. The *orbit* of a model $M$ under a symmetry group $\Theta$ is the set of models that can be obtained by applying any of the symmetries in $\Theta$. A symmetry group divides the space of models into disjoint sets, where the models in an orbit either all satisfy or all do not satisfy the theory. The idea of symmetry-breaking is to add clauses to $\mathcal{T}$ rule out many of the models, but are guaranteed to *not* rule out at least one model in each orbit. Note that symmetry-breaking preserves satisfiability of a theory.

Symmetries can be found in CNF theories using a reduction to graph isomorphism, a problem that is thought to require super-polynomial time in the worst case, but which can often be efficiently solved in practice [18]. The added clauses are called *symmetry-breaking predicates* (SBPs). If we place a fixed order on the atoms of theory, then a model can be associated with a binary number, where the $i$-th digit, 0 or 1, specifies the value of the $i$-th atom, false or true. *Lex-leader* SBPs rule out models that are not the lexicographically-smallest members of their orbits. The formulation below is equivalent to the lex-leader SBP given by Crawford et. al. in [7]:

$$SBP(\theta) = \bigwedge_{1 \le i \le n} \Big( \bigwedge_{1 \le j < i} v_j \Leftrightarrow \theta(v_j) \Big) \Rightarrow v_i \Rightarrow \theta(v_i) \tag{1}$$

where $v_i$ is the $i$th variable in the ordering of $n$ variables, and $\theta$ is a symmetry over variables. Even though graph isomorphism is relatively fast in practice, a theory may have exponentially many symmetries. Therefore, breaking all the symmetries is often impractical, though partial symmetry-breaking is still useful. It is possible to devise new SBPs that can break exponentially more symmetries than the standard form described above; we do so in Section 5.2.

**Relational Theories**   We define a *relational theory* as a tuple $\mathcal{T} = (F, W, \mathcal{E})$, where $F$ is a set of first-order formulas, $W$ a mapping of predicates and negated predicates to strictly positive real numbers (weights), and $\mathcal{E}$ is a set of evidence. We restrict the formulas in $F$ to be built from predicates, variables, quantifiers, and logical connectives, but no constants or function symbols. $\mathcal{E}$ is a set of ground literals; that is, literals built from predicates and constant symbols. The predicate arguments and constant symbols are typed. Universal and existential quantification is over the set of the theory's constants $\mathcal{D}$ (*i.e.* the constants that appear in its evidence). Any constants not appearing explicitly in the evidence can be incorporated by introducing a unary predicate for each constant type and adding the groundings for those constants to the evidence. Any formula containing a constant can be made constant-free, by introducing a new unary predicate for each constant, and then including that predicate applied to that constant in the evidence. A ground theory can be seen as a special case of a relational theory where each predicate is argument free.

We define the weight of a positive ground literal $P(C_1, \ldots, C_k)$ of a theory as $W(P)$, and the weight of negative ground literal $\neg P(C_1, \ldots, C_k)$ as $W(\neg P)$. In other words, all positive groundings of a literal have the same weight, as do all negative groundings. The weight of model $M$ with respect to a theory $(F, W, \mathcal{E})$ is 0 if $M$ fails to satisfy any part of $F$ or $\mathcal{E}$; otherwise, it is the product of the weights of the ground atoms that are true in $M$. *Maximum weighted model-finding* is the task for finding a model of maximum weight with respect to $\mathcal{T}$. A relational theory can be taken to define a probability distribution over the set of models, where the probability of model is proportional to its

weight. Maximum weighted model-finding thus computes MAP (most probable explanation) for a given theory. Ordinary satisfiability corresponds to the case where $W$ simply sets the weights of all literals to 1.

Languages such as Markov Logic [12] use an alternative representation and specify real-valued weights on formulas rather than positive weights on predicates and their negations. The MAP problem can be formulated as the *weighted-MaxSAT* problem, *i.e.* finding a model maximizing the sum of the weights of satisfied clauses. This can be translated to our notation by introducing a new predicate for each original formula, whose arguments are the free variables in the original formula. $F$ asserts that the predicate is equivalent to the original formula, and $W$ asserts that the weight of the new predicate is $e$ raised to the weight of the original formula. Solving weighted MaxSAT in the alternate representation is thus identical to solving maximum weighted model-finding in the translated theory. For the rest of the discussion in this paper, we will assume that the theory is specified with weights on predicates (and their negations).

## 3   Related Work

Our work has connections to research in both the machine learning and constraint-satisfaction research communities. Most research in statistical-relational machine learning has concentrated on modifying or creating novel probabilistic inference algorithms to exploit symmetries, as opposed to symmetry-breaking's solver-independent approach. Developments include lifted versions of variable elimination [27, 8], message passing [29, 30, 23], and DPLL [14]. Our approach of defining symmetries using group theory and detecting them by graph isomorphism is shared by Bui *et al.*'s work on lifted variational inference [5] and Apsel *et al.*'s work on cluster signatures [2]. Bui notes that symmetry groups can be defined on the basis of *unobserved* constants in the domain, while we have developed methods to explicitly find symmetries among constants that do appear in the evidence. Niepert also gives a group-theoretic formalism of symmetries in relational theories [24, 25], applying them to MCMC methods. Another line of work is to make use of problem transformations. First-order knowledge compilation [10, 11] transforms a relational problem into a form for which MAP and marginal inference is tractable. This is a much more extensive and computationally complex transformation than symmetry-breaking. Mladenov *et al.* [22] propose an approach to translate a linear program over a relational domain into an equivalent lifted linear program based on message passing computations, which can then be solved using any off-the-shelf LP solver. Recent work on MAP inference in Markov Logic has identified special cases where a relational formula can be transformed by replacing a quantified formula with a *single* grounding of the formula [21].

Relatively little work in SRL has explicitly examined the role of evidence, separate from the first-order part of a theory, on symmetries. One exception is [31], which presents a heuristic method for approximating an evidence set in order to increase the number of symmetries it induces. Bui et al. [4] consider the case of a theory plus evidence pair, where the theory has symmetries that are broken by the evidence. They show that if the evidence is soft and consists only of unary predicates, then lifting based on the theory followed by incorporation of evidence enables polynomial time inference. Extending the results of [4], Van Den Broeck and Darwiche [9] show that dealing with binary evidence is NP-hard in general but can be done efficiently if there is a corresponding low rank Boolean matrix factorization. They also propose approximation schemes based on this idea.

We briefly touch upon the extensive literature that has grown around the use of symmetries in constraint satisfaction. Symmetry detection has been based either on graph isomorphism on propositional theories as in the original work by by Crawford *et. al* [7]; by interchangeability of row and/or columns in CSPs specified in matrix form [20]; by checking for other special cases of geometric symmetries [28], or by determining that domain elements for a variable are exchangeable [3]. (The last is a special case of our term equivalent symmetries.) Researchers have suggested symmetry-aware modifications to backtracking CSP solvers for variable selection, branch pruning, and no-good learning [20, 13]. A recent survey of symmetry breaking for CSP [32] described alternatives to the lex-leader formulation of SBPs, including one based on Gray codes.

## 4   Symmetries in Relational Theories

In this section, we will formally introduce the notion of symmetries over relational theories and give efficient algorithms to find them. Symmetries of a relational theory can be defined in terms of symmetries over the corresponding ground theory.

**Definition 4.1.** Let $\mathcal{T}$ denote a relational theory. Let the $\mathcal{T}^G$ denote the theory obtained by grounding the formulas in $\mathcal{T}$. Let $L$ denote the set of (ground) literals in $\mathcal{T}^G$. We say that a permutation $\theta$ of the set $L$ is a symmetry of the relational theory $\mathcal{T}$ if $\theta$ maps the ground theory $\mathcal{T}^G$ back to itself *i.e.* $\theta(\mathcal{T}^G) = \mathcal{T}^G$. We denote the action of $\theta$ on the original theory as $\theta(\mathcal{T}) = \mathcal{T}$.

A straightforward way to find symmetries over a relational theory $\mathcal{T}$ is to first map it to corresponding ground theory $\mathcal{T}^G$ and then find symmetries over it using reduction to graph isomorphism. The complexity of finding symmetries in this way is the same as that of graph isomorphism, which is believed to be worst-case super-polynomial. Further, the number of symmetries found is potentially exponential in the number of ground literals. This is particularly significant for relational theories since the number of ground literals itself is exponential in the highest predicate arity. Computing symmetries at the ground level can therefore be prohibitively expensive for theories with high predicate arity and many constants. In our work below, we exploit the underlying template structure of the relational theory to directly generate symmetries based on the evidence.

## 4.1 Term Symmetries

We introduce the notion of symmetries defined over terms (constants) appearing in a theory $\mathcal{T}$, called *term symmetries*.

**Definition 4.2.** Let $\mathcal{T}$ be a relational theory. Let $\mathcal{D}$ be the set of constants appearing in the theory. Then, a permutation $\theta$ over the term set $\mathcal{D}$ is said to be a *term symmetry* with respect to evidence $\mathcal{E}$ if application of $\theta$ on the terms appearing in $\mathcal{E}$, denoted by $\theta(\mathcal{E})$, maps $\mathcal{E}$ back to itself. We will also refer to $\theta$ as an evidence symmetry for the set $\mathcal{E}$.

The problem of finding term symmetries can be reduced to colored graph isomorphism. We construct a graph $G$ as follows: for each predicate $P$ and it's negation, $G$ has a node and a unique color is assigned to every such node. $G$ also has a unique color for each type in the domain. There is a node for every term which takes the color of its type. We call an ordered list of terms an *argument list*, *e.g.*, given the literal $P(C_1, C_2)$, where $C_1, C_2 \in D$, the argument list is $(C_1, C_2)$. The type of an argument list is simply the cross-product of the types of the terms appearing in it. $G$ has a node for every argument list appearing in the evidence, which takes the color of its type.

For every evidence literal, there is an edge between the predicate node (or its negation) and the corresponding argument list node. There is also an edge between the argument list node and each of the terms appearing in the list. Thus, for the previous example, an edge will be placed between the node for $P$ and the node for $(C_1, C_2)$, as well as an edge each between $(C_1, C_2)$ and the nodes for $C_1$ and $C_2$.

Any automorphism of $G$ will map (negated) predicate nodes to themselves and terms will be mapped in a manner that their association with the corresponding predicate node in the evidence is preserved. Hence, automorphisms of $G$ will correspond to term symmetries in evidence $\mathcal{E}$. Next, we will establish a relationship between permutation of terms in the evidence to the permutations of literals in the ground theory.

**Definition 4.3.** Let $\mathcal{T}$ be a relational theory. Let $\mathcal{E}$ be the evidence set and let $\mathcal{D}$ be the set of terms appearing in $\mathcal{E}$. Given a permutation $\theta$ of the terms in the set $\mathcal{D}$, we associate a corresponding permutation $\theta^{\mathcal{T}}$ over the ground literals of the form $P(C_1, C_2, \ldots, C_k)$ in $T$, such that $\theta^{\mathcal{T}}(P(C_1, C_2, \ldots, C_k)) = P(\theta(C_1), \theta(C_2), \ldots, \theta(C_k))$ (and similarly for negated literals).

We can now associate a symmetry $\theta$ over the terms with a symmetry of the theory $\mathcal{T}$. The following lemma is proven in the supplementary material:

**Lemma 4.1.** Let $\mathcal{T}$ be a relational theory. Let $\mathcal{E}$ denote the evidence set. Let $\mathcal{D}$ be the set of terms appearing in $\mathcal{E}$. If $\theta$ is term symmetry for the evidence $\mathcal{E}$, then, the associated theory permutation $\theta^{\mathcal{T}}$ is also a symmetry of $\mathcal{T}$.

In order to find the term symmetries, we can resort to solving a graph isomorphism problem of size $O(|\mathcal{E}|)$, where $|\mathcal{E}|$ is the number of literals in the evidence. Directly finding symmetries over the ground literals requires solving a problem of size $O(|\mathcal{D}|^k)$, $|\mathcal{D}|$ being the set of constants and $k$ being the highest predicate arity. In the worst case where everything is fully observed, $|\mathcal{E}| = O(|\mathcal{D}|^k)$, but in practice it is much smaller. Next, we present an important subclass of term symmetries, called *term equivalent symmetries*, which capture a wide subset of all the symmetries present in the theory, and can be efficiently detected and broken.

## 4.2 Term Equivalent Symmetries

A *term equivalent* symmetry is a set of term symmetries that divides the set of terms into equivalence classes such that any permutation which maps terms within the same equivalence class is a symmetry of the evidence set. Let $Z = \{Z_1, Z_2, \ldots, Z_m\}$ denote a partition of the term set $\mathcal{D}$ into $m$ disjoint subsets. Given a partition $Z$, we say that two terms are *term equivalent* (with respect to $Z$) if they occur in the same component of $Z$. We define a partition preserving permutation as follows.

**Definition 4.4.** Given a set of terms $\mathcal{D}$ and its disjoint partition $Z$, we say that a permutation $\theta$ of the terms in $\mathcal{D}$ is a partition preserving permutation of $\mathcal{D}$ with respect to the partition $Z$ if $\forall C_j, C_k \in \mathcal{D}, \theta(C_j) = C_k$ implies that $\exists Z_i \in Z$ st $C_j, C_k \in Z_i$. In other words, $\theta$ is partition preserving if it permutes terms within the same component of $Z$.

The set of all partition preserving permutations (with respect to a partition $Z$) forms a group. We will denote this group by $\Theta^Z$. It is easy to see that $\Theta^Z$ divides the set of terms in equivalence classes. Next, we define the notion of term equivalent symmetries.

**Definition 4.5.** Let $\mathcal{T}$ be a relational theory and $\mathcal{E}$ denote the evidence set. Let $\mathcal{D}$ be the set of terms in $\mathcal{E}$ and $Z$ be a disjoint partition of terms in $\mathcal{D}$. Then, given the partition preserving permutation $\Theta^Z$, we say that $\Theta^Z$ is a term equivalent symmetry group of $\mathcal{D}$, if $\forall \theta \in \Theta^Z$, $\theta$ is a symmetry of $\mathcal{E}$. We will refer to each symmetry $\theta \in \Theta^Z$ as a term equivalent symmetry of $\mathcal{E}$.

A partition $Z$ of term set $\mathcal{D}$ is a term equivalent partition if the partition preserving group $\Theta^Z$ is a symmetry group of $\mathcal{D}$. We refer to each partition element $Z_i$ as a term equivalent subset. The term equivalent symmetry group can be thought of as a set of symmetry subgroups $\Theta^{Z_i}$'s, one for each term subset $Z_i$, such that, $Z_i$ allows for all possible permutations of terms within the set $Z_i$ and defines an identity mapping for terms in other subsets. Note that the size of term equivalent symmetry group is given by $\Pi_{i=1}^m |Z_i|!$. Despite its large size, it can be very efficiently represented by simply storing the partition $Z$ over the term set $\mathcal{D}$. Note that this formulation works equally well for typed as well untyped theories; in a typed theory no two terms of differing types can appear in the same subset. Next, we will look at an efficient algorithm for finding a partition $Z$ which corresponds to a term equivalent symmetry group over $\mathcal{D}$.

Let the evidence be given by $\mathcal{E} = \{l_1, l_2, \ldots, l_k\}$, where each $l_i$ is a ground literal. Intuitively, two terms are *term equivalent* if they occur in exactly the same context in the evidence. For example, if evidence for constant $A$ is $\{P_1(A, X), P_2(A, Y, A)\}$, then the context for term $A$ is $P_1(*, X), P_2(*, Y, *)$. Note that here the positions where $A$ occurred in the evidence has been marked by a $*$. Any other term sharing the same context would be term equivalent to $A$. To find the set of all the equivalent terms, we first compute the context for each term. Then, we sort each context based on some lexicographic order defined over predicate symbols and term symbols. Once the context has been sorted, we can simply hash the context for each term and put those which have the same context in the same equivalence class. If the evidence size is given by $|\mathcal{E}| = M$ and number of terms in evidence is $n$, then, the above procedure will take $O(nM \log(M))$ time. The $M \log(M)$ factor is for sorting the context for a single term. Hashing the sorted term takes constant time. This is done for each term, hence the factor of $n$. See the supplement for more details and an example.

## 5 Breaking Symmetries Over Terms

In this section, we provide the SBP formulation for term as well as term equivalent symmetries.

### 5.1 Breaking Term Symmetries

Consider a relational theory $\mathcal{T}$ that has term symmetries $\{\theta_1, \ldots, \theta_k\}$. Fix an ordering $P_1, \ldots, P_r$ over the predicates, an ordering over the predicate positions, and an ordering $C_1, \ldots, C_{|\mathcal{D}|}$ over the terms. If $\mathcal{T}$ is typed, fix an ordering over types and an ordering over the terms within each type. This induces a straightforward ordering over the ground atoms of the theory $G_1, \ldots, G_n$.

Let $\theta$ be a term symmetry. Consider the following SBP (based on Equation 1) to break $\theta$:

$$SBP(\theta) = \bigwedge_{1 \leq i \leq n} \Big( \bigwedge_{1 \leq j < i} G_j \Leftrightarrow \theta(G_j) \Big) \Rightarrow G_i \Rightarrow \theta(G_i)$$

**Theorem 5.1.** If $\mathcal{T}$ is weighted, the max model for $\mathcal{T} \wedge SBP(\theta)$ is a max model for $\mathcal{T}$, and it has the same weight in both theories.

The proof of this theorem follows from [7]. Essentially, an ordering is placed on the atoms of the theory, which induces an ordering on the models. The SBP constraints ensure that if a set of models are in the same orbit, *i.e.* symmetric under $\theta$, then only the first of those models in the ordering is admitted.

## 5.2 Breaking Term Equivalent Symmetries

Let $Z = \{Z_1, \ldots, Z_{|Z|}\}$ be the term equivalent partitioning over the terms. Let $\theta_{j,k}^i$ be the term symmetry that swaps $C_j$ and $C_k$, the $j$th and $k$th constants in an ordering over the term equivalent subset $Z_i$, and maps everything else to identity. We next show how to break exponentially many symmetries that respect the term equivalent partition $Z$, using a linear-sized SBP. Consider the following SBP (CSBP stands for Composite SBP):

$$CSBP(Z) = \bigwedge_{i=1}^{|Z|} \bigwedge_{j=1}^{|Z_i|-1} SBP(\theta_{j,j+1}^i)$$

For each term (equivalent) symmetry of the form $\theta_{j,j+1}^i$, we apply the SBP formulation from Section 5.1 to break it. The formulation is cleverly choosing which symmetries to explicitly break, so that exponentially many symmetries are broken. The inner conjunction iterates over all of the terms in a term equivalent class. An ordering is placed on the terms, and the only symmetries that are explicitly broken are those that swap two adjacent terms and map everything else to identity. All of the adjacent pairs of terms in the ordering are broken in this way, with a linear number of calls to SBP. As we show below, this excludes $\Omega(2^{|Z_i|})$ models from the search space, while preserving at least one model in each orbit. The outer conjunction iterates over the different term equivalent subsets, breaking each one of them in turn. The following theorem states this in formal terms (see the supplement for a proof):

**Theorem 5.2.** $CSBP(Z)$ removes $\Omega(\sum_{i=1}^{|Z|} 2^{|Z_i|})$ models from the search space while preserving at least one model in each orbit induced by the term equivalent symmetry group $\Theta^Z$.

**Corollary 5.1.** If $\mathcal{T}$ is weighted, the max model for $\mathcal{T} \wedge CSBP(Z)$ is a max model for $\mathcal{T}$, and it has the same weight in both theories.

# 6 Experiments

We pitted our term and term equivalent SBP formulations against other inference systems to demonstrate their efficacy. For each domain we tested on, we compared the following algorithms: (1) Vanilla: running an exact MaxSAT solver on the grounded instance (2) Shatter: running an exact MaxSAT solver on the grounded instance with SBPs added by the Shatter tool [1], (3) Term: running an exact MaxSAT solver on the grounded instance with SBPs added by our term symmetry detection algorithm, (4) Tequiv: running an exact MaxSAT solver on the grounded instance with SBPs added by our term equivalent symmetry detection algorithm (5) RockIt: running RockIt [26] on the MLN, (6) PTP: running the PTP algorithm [14] for marginal inference [1] on the MLN and (7) MWS: running the approximate MaxWalkSAT algorithm used by Alchemy-2[2] on the grounded instance. These algorithms were chosen so that our methods would be compared against a variety of other techniques: ground techniques including regular MaxSAT (Vanilla), propositional symmetry-breaking (Shatter), local search (MWS), and lifted techniques including cutting planes (RockIt) and lifted rules (PTP). It should be noted that all the algorithms except MWS and RockIt are exact. RockIt solves an LP relaxation but it always gave us an exact solution (whenever it could run). Therefore, we report the solution quality only for MWS. All the experiments were run on a system with a 2.2GHz Xeon processor and 62GB of memory.

The algorithms for Term and Tequiv were implemented as a pre-processing step before running a MaxSAT solver. The MWS runs were allowed a varying number of flips and up to five restarts. For MWS, the reported results are with equal probability (0.5) of making a random or a greedy move; other settings were tried, and the results were similar. Both PTP and RockIt were run using the default setting of their parameters. We experimented on two relational domains: the classic pigeonhole problem (PHP) with two different variants, and an Advisor domain (details below).

For ground instances that required an exact MaxSAT solver, we experimented with three different solvers: MiniMaxSAT [15], Open-WBO [19], and Sat4j [17]. This was done because different solvers and heuristics tend to work better on different application domains. We found that for PHP (variant 1) Sat4j worked best, for PHP (variant 2) Open-WBO was the best and for Advisor Mini-MaxSAT worked the best. We used the best MaxSAT solver for each domain.

**PHP:** In the PHP problem, we are given $n$ pigeons and $n-1$ holes. The goal is to fit each pigeon in a unique hole (the problem is unsatisfiable). We tried two variants of this problem. In the first variant, there are hard constraints that ensure that every hole has at most one pigeon and every pigeon is in at most one hole. There is a soft constraint that states that every pigeon is in every hole. The goal is to find a max-model of the problem. The comparison with various algorithms is given in Table 1a. Vanilla times out except for the smallest instance. PTP times out on all instances. Our algorithms consistently outperform RockIt but are not able to outperform shatter (except for smaller instances where they are marginally better). MWS was run on the largest instance (n=60) and results are shown in Table 1d. MWS is unable to find the optimal solution after 10 million flips.

In the second PHP variant, there are hard constraints that ensure that every pigeon is in precisely one hole. There is a soft constraint that each hole has at most one pigeon. The comparison with various algorithms is given in in Table 1b. Vanilla times out except for the two smallest instances, and PTP times out for all the instances. Our systems consistently outperform RockIt, and outperform shatter for the larger instances. Tequiv outperforms Term because the detection step is much faster. MWS (Table 1d) is able to find the optimal within 100,000 flips and is significantly faster than all other algorithms. Though MWS does better on this variant of PHP, it fails badly on the first one. Further, there is no way for MWS to know if it has reached the optimal unlike our (exact) approaches.

**Advisor Domain:** The advisor-advisee domain has three types: student, prof, and a research area. There are predicates to indicate student/prof interest in an area and an advisor relationship between students and profs. The theory has hard constraints that ensure that a) each student has precisely one advisor b) students and their advisors share an interest. It also has a small negatively-weighted (soft) constraint saying that a prof advises two different students. The interests of all students and profs are fully observed. Students have one interest, profs have one or more (chosen randomly). There is at least one prof interested in each area. The results for this domain are given in Table 1c gives the comparisons. Vanilla is able to run only on the two smallest instances. PTP times out on every instance as before. Our algorithms outpeform shatter but is outperformed by RockIt. MWS was run on the two larger sized instances (see Table 1c) and it is able to outperform our system.

In order to make sure that poor performance of PTP is not due to evidence, we modified the PHP formulation to work with a no evidence formulation. This did not help PTP either. Based on these results, there is no clear winner among the algorithms which performs the best on all of these domains. Among all the algorithms, Term, Tequiv and Shatter are the only ones which could run to completion on all the instances that we tried. Among these, our algorithms won on PHP variant 2 (large instances) and the Advisor domain. Tequiv outperformed Term on PHP variant 2 (large instances), and they performed similarly in the Advisor domain.

In the PHP variants, all of the pigeons are in one term equivalent symmetry group, and all of the holes are in another. The PHP is of special interest, because pigeonhole structure was shown to be present in hard random SAT formulas with very high probability [6]. In the advisor domain, there is a term equivalent symmetry group of students for each area, and term equivalent symmetry group of professors for each set of areas.

# 7 Conclusion and Future Work

In this work, we have provided the theoretical foundation for using symmetry-breaking techniques from satisfiability testing for MAP inference in weighted relational theories. We have presented the class of term symmetries, and a subclass of term equivalent symmetries, both of which are found in the evidence given with the theory. We have presented ways to detect and break both these classes. For the class of term equivalent symmetries, the detection method runs in low-polynomial time, and the number of SBPs required to break the symmetries is linear in the domain size. the algorithms presented, and compared them against other systems on a set of relational domains. Future work includes carefully characterizing the cases where our algorithms perform better than other systems, experimenting on additional real-world domains, comparing with more lifted inference approaches, and the application of similar techniques to the problem of marginal inference.

Table 1: Experiments with Lifted Symmetries for MAP Inference

(a) Soft Pigeon Hole Domain — Variant 1

| #Pigeon | Vanilla | Shatter | Tequiv | Term | RockIt | PTP |
|---|---|---|---|---|---|---|
| 5 | 0.246 | 0.01 + 0.39 | 0.01 + 0.24 | 0.03 + 0.26 | 1.15 | TO |
| 10 | TO | 0.02 + 0.88 | 0.04 + 0.74 | 0.09 + 0.68 | 1.47 | TO |
| 15 | TO | 0.08 + 1.85 | 0.13 + 1.37 | 0.21 + 1.27 | 2.46 | TO |
| 20 | TO | 0.22 + 3.1 | 0.36 + 3.9 | 0.51 + 3.7 | 2.70 | TO |
| 30 | TO | 1.6 + 12.7 | 1.4 + 20 | 2.2 + 14 | 3.88 | TO |
| 40 | TO | 7 + 72 | 4.06 + 93.3 | 6.30 + 88.3 | F | TO |
| 60 | TO | 54 + 889 | 18 + 1128 | 28 + 1128 | F | TO |

(b) Soft Pigeon Hole Domain — Variant 2

| #Pigeon | Vanilla | Shatter | Tequiv | Term | RockIt | PTP |
|---|---|---|---|---|---|---|
| 5 | 0.002 | 0.003 + 0.001 | 1.28 + 0.002 | 0.04 + 0.001 | 1.1 | TO |
| 10 | 37.44 | 0.01 + 0.006 | 1.19 + 0.004 | 0.07 + 0.005 | 6.6 | TO |
| 15 | TO | 0.06 + 0.01 | 1.29 + 0.01 | 0.19 + 0.01 | TO | TO |
| 40 | TO | 3.7 + 0.59 | 4.9 + 0.17 | 6.2 + 0.21 | F | TO |
| 80 | TO | 115 + 8.36 | 50 + 1.7 | 86 + 2.9 | F | TO |
| 125 | TO | 1090 + 20.98 | 269 + 10 | 486 + 18 | F | TO |

(c) Advisor Domain

| #Prof–#Student–#Area | Vanilla | Shatter | Tequiv | Term | RockIt | PTP |
|---|---|---|---|---|---|---|
| 4–12–2 | 2.3 | 0.01 + 0.10 | 0.02 + 0.01 | 0.11 + 0.01 | 1.25 | TO |
| 4–16–2 | 168 | 0.01 + 0.17 | 0.03 + 0.05 | 0.08 + 0.04 | 2.06 | TO |
| 6–18–3 | TO | 0.02 + 0.90 | 0.05 + 0.84 | 0.09 + 0.80 | 6.56 | TO |
| 6–24–3 | TO | 0.03 + 105 | 0.07 + 45 | 0.15 + 47 | 1.47 | TO |
| 8–24–4 | TO | 0.07 + 62 | 0.10 + 56 | 0.16 + 55 | 13.4 | TO |

(d) MaxWalkSAT Results

| Problem | Size | Optimal | 100,000 | | 1,000,000 | | 10,000,000 | |
|---|---|---|---|---|---|---|---|---|
| | | | best | time | best | time | best | time |
| pigeon1 | 60 | 3481 | 3496 | 4.4 | 3494 | 39 | 3491 | 383 |
| pigeon2 | 125 | 1 | 1 | 3.8 | — | — | — | — |
| advisor | 6–18–3 | 2094 | 2094 | 0.4 | — | — | — | — |
| advisor | 8–24–4 | 3552 | 3552 | 0.5 | — | — | — | — |

All times are given in seconds. "TO" indicates the program timed out (30min). "F" indicates the program failed without timeout (instance too large). The vanilla column is the time to find the max model of the ground theory. The first three subtables give results for exact algorithms. The Shatter, Term, and Tequiv columns are the same with propositional, term and term equivalent symmetries broken, respectively. These columns are given as $x + y$, where $x$ is the time to detect and break the symmetries, and $y$ is the time to find the max model. The RockIt column is the time it takes RockIt to find the max model. The lifted PTP algorithm was run on every instance, and timed out on each instance. The last table gives the results for the approximate algorithm MaxWalkSAT, giving the time it takes to perform a certain number of iterations, as well as the comparison between the best model found on that run and the optimal model.

## Footnotes

[1] The publicly available implementation of PTP only supports marginal inference. Though marginal inference is somewhat harder problem than MAP, PTP failed to run even on the smallest of instances.

[2] https://code.google.com/p/alchemy-2/

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
