[Supplementary Material · supplement.pdf]

# Lifted Symmetry Detection and Breaking for MAP Inference — Supplemental Material

**Tim Kopp**
University of Rochester
tkopp@cs.rochester.edu

**Parag Singla**
Indian Institute of Technology
parags@cse.iitd.ac.in

**Henry Kautz**
University of Rochester
kautz@cs.rochester.edu

## 4.1 Term Symmetries

**Lemma 4.1.** *Let $\mathcal{T}$ be a relational theory. Let $\mathcal{E}$ denote the evidence set. Let $\mathcal{D}$ be the set of terms in the domain. If $\theta$ is an evidence symmetry of $\mathcal{E}$, then, the associated theory permutation $\theta(T)$ is also a symmetry of $\mathcal{T}$.*

*Proof.* Consider $\mathcal{T}^g$, the result of grounding $\mathcal{T}$. In a theory with evidence, the ground theory is the result of grounding every constraint, and adding the (already grounded) evidence to that ground theory: $\mathcal{T}^g \cup \mathcal{E}$.

Let $\theta$ be an evidence permutation of $\mathcal{E}$. By our assumption, we know that $\theta$ is a symmetry of $\mathcal{E}$. We will show that $\theta$ is a symmetry of $\mathcal{T}^g$, which suffices to prove that $\theta$ is a symmetry of $\mathcal{T}^g \cup \mathcal{E}$.

Consider a single first order constraint $\varphi \in \mathcal{T}$. The grounding of that constraint $\varphi^g$ is the set of *all possible* groundings of $\varphi$ over $\mathcal{D}$. Since we assume $\mathcal{T}$ has no constants, this means that every combination of assignments of terms to variables in $\varphi$ is included in the grounding. Now consider an arbitrary permutation of terms (that respects types, as $\theta$ is guaranteed to). Since the grounding of $\varphi$ includes every combination of (type-aware) assignments of terms to variables, and no constants are included, it is clear that $\theta(\varphi^g) = \varphi^g$.

Since $\theta$ is a symmetry for arbitrary $\varphi^g \in \mathcal{T}^g$, and since the constraints are independent, $\theta$ must be a symmetry for $\varphi_1^g \cup \varphi_2^g \ldots \varphi_k^g = \mathcal{T}^g$. Since $\theta$ is a symmetry of $\mathcal{T}^g$ and of $\mathcal{E}$, and $\mathcal{E}$ can be viewed as a set of ground hard unit clauses, $\theta$ must be a symmetry of $\mathcal{T}^g \cup \mathcal{E}$, and therefore of $\mathcal{T}$. $\qquad\square$

## 4.2 Term Equivalent Symmetries

**Theorem.** *A set of terms $\mathcal{S} \subseteq \mathcal{D}$ is term equivalent in a relational theory $\mathcal{T}$ if and only if every $C \in \mathcal{S}$ has the same context.*

*Proof.* Suppose $\mathcal{S} \subseteq \mathcal{D}$ is term equivalent in $\mathcal{T}$. By definition, this means that there is a set of symmetries $\Theta$, that partitions $\mathcal{D}$ into $Z = Z_1, \ldots, Z_{|Z|}$, $\mathcal{S} \subseteq Z_i$. Suppose that two terms $C_1$ and $C_2$ are elements of $\mathcal{S}$, but have different contexts. If the contexts are different, then there is a literal position $l$ that appears in one term's context but not the other's. Wlog, suppose $l$ is in the context of $C_1$ but not $C_2$. Let $l_{C_1}$ be the literal that results from placing $C_1$ in the literal position $l$. Now consider the symmetry $\theta$ that swaps $C_1$ and $C_2$, and maps everything else to identity. By definition, $\theta \in \Theta$. By definition of symmetry, $\mathcal{T} = \theta(\mathcal{T})$. However, if $l$ is not a literal position for $C_2$, then $\theta(l_{C_1}) \notin E$, and therefore $\mathcal{T} \neq \theta(\mathcal{T})$. Contradiction. Therefore $C_1$ and $C_2$ have the same context. Since this can be done for arbitrary $C_1, C_2 \in \mathcal{S}$, all terms in $S$ have the same context.

Suppose every $C \in \mathcal{S} \subseteq \mathcal{D}$ has the same context. Consider the term equivalent symmetry $\Theta$, which induces disjoint partitions $Z = Z_1, \ldots, Z_{|Z|}$. Consider any two $C_1, C_2 \in \mathcal{S}$. Suppose wlog $C_1 \in Z_1$. Suppose $C_1$ and $C_2$ are not term equivalent. By definition, $C_2 \notin Z_1$. By definition of term equivalent symmetry, the permutation $\theta$ that swaps $C_1$ and $C_2$, while mapping other terms to identity, must not by a symmetry, that is, $\theta(\mathcal{T}) \neq \mathcal{T}$. However, since the constraints contain no

terms, and $C_1$ and $C_2$ have the same contexts Therefore $C_2 \in Z_1$. Since this can be done for any arbitrary pair of terms in $\mathcal{S}$, $\mathcal{S} \subseteq Z_1$, and $\mathcal{S}$ is term equivalent. $\qquad\square$

**Example** Let the *context* of a term be a representation of the set of literal positions in which is appears in $\mathcal{E}$. Consider the example evidence below:

$$P_1(C_1, C_2, C_1), \quad P_1(C_3, C_2, C_3), \quad P_2(C_1), \quad \neg P_2(C_2),$$
$$P_2(C_3), \quad P_3(C_1, C_2), \quad P_3(C_3, C_2), \quad P_3(C_2, C_4)$$

The contexts for the constants appearing in that evidence are:

$$
\begin{aligned}
C_1 : &\quad P_1(*, C_2, *), P_2(*), P_3(*, C_2)\\
C_2 : &\quad P_1(C_1, *, C_1), P_1(C_3, *, C_3), \neg P_2(*), P_3(C_1, *), P_3(C_3, *), P_3(*, C_4)\\
C_3 : &\quad P_1(*, C_2, *), P_2(*), P_3(*, C_2)\\
C_4 : &\quad P_3(C_2, *)
\end{aligned}
$$

If there happened to be an additional term $C_5$ in the domain, its context would be empty.

## 5.2 Breaking Term Equivalent Symmetries

Let $Z = \{Z_1, \dots, Z_{|Z|}\}$ be the term equivalent partitioning over the terms. Let $\theta^i_{j,k}$ be the term symmetry that swaps $C_j$ and $C_k$, the $j$th and $k$th constants in an ordering over the term equivalent partition $Z_i$, and maps everything else to identity. We show how to break exponentially many symmetries that respect the term equivalent partition $Z$. Consider the following SBP (CSBP stands for Composite SBP):

$$CSBP(Z) = \bigwedge_{i=1}^{|Z|} \bigwedge_{j=1}^{|Z_i|-1} SBP(\theta^i_{j,j+1})$$

**Theorem 5.2.** $CSBP(Z)$ *removes* $\Omega(\sum_{i=1}^{|Z|} 2^{|Z_i|})$ *models from the search space while preserving at least one model in each orbit induced by the term equivalent symmetry group* $\Theta^Z$.

*Proof.* We first show that the inner conjunction, which iterates over the terms in $Z_i$, removes at least $2^{|Z_i|} - (|Z_i| + 1)$ models from the search space. Consider a predicate $P$ that contains an argument of the type of $Z_i$, and possibly arguments of other types. Suppose $P$ is the first predicate in the ordering that contains a variable of that type. Further, for now, let us assume that $P$ contains a single argument of the type of $Z_i$ (we will relax it later).

Since $P$ is the first predicate in the ordering that contains an argument of the type of $Z_i$, the inner conjunction ofs CSBP for $Z_i$ will contain the following clauses:

$$
\begin{aligned}
P(\dots, C_1^{Z_i}, \dots) &\Rightarrow P(\dots, C_2^{Z_i}, \dots)\\
P(\dots, C_2^{Z_i}, \dots) &\Rightarrow P(\dots, C_3^{Z_i}, \dots)\\
&\vdots\\
P(\dots, C_{|Z_i|-1}^{Z_i}, \dots) &\Rightarrow P(\dots, C_{|Z_i|}^{Z_i}, \dots)
\end{aligned}
$$

The conjunction of these clauses ensures that of the groundings of $P$ that contain a term in $Z_i$, there is precisely one model admitted in which $k$ groundings of $P$ are true, for each value of $k$ ranging from 0 to $|Z_i|$. Namely, these are the models in which only the last $k$ groundings in the ordering above are true. There are $2^{|Z_i|}$ assignments to the groundings of $P$ that contain a term in $Z_i$, and all but $|Z_i| + 1$ are eliminated. Therefore, the number of models that are removed from the search space is $2^{|Z_i|} - (|Z_i| + 1)$.

This represents a lower bound on the number of models eliminated from the search space. If the theory contains more than one predicate containing arguments of the type of $Z_i$, more models will be eliminated. Furthermore, if the first predicate in the ordering contains more than one variable of

the desired type, then at least that many, but potentially more are removed (note that we can still apply the above reasoning to the first argument with type of $Z_i$ in $P$).

These models were eliminated by the inner conjunction for the term equivalent subset $Z_i$. CSBP iterates over each of the term equivalent subsets. We can therefore apply the same reasoning over each such $Z_i$. Further, the predicate groundings considered in each case are disjoint from each other (since we are considering different term equivalent subsets). This results in a lower bound of $\Omega(\sum_{i=1}^{|Z|} 2^{|Z_i|})$ on the number of models that are eliminated from the search space.

Now we must show that at least one model is preserved in each orbit of the term equivalent symmetry group being considered. By the theory of symmetry breaking (see Section 2), each individual call to an SBP preserves at least one model in each orbit. Furthermore, the lex leader of each orbit will definitely be preserved under the given ordering. Therefore, since each of the calls to an SBP use the same ordering, no call to an SBP will remove the lex leader of an orbit from the search space. Therefore, the conjunctions of the calls to SBPs will preserve at least the lex leader in each orbit. $\quad\square$