[Reviews · NeurIPS 2015]

Submitted by Assigned_Reviewer_1

SUMMARY

The work concerns symmetries in relational models, which are

permutations of the set of atoms (i.e. predicates applied to

constants) that do not change a relational theory.

A model of

a theory is a truth-assignment for each atom that makes the

theory true.

A symmetry of a theory can therefore permute the

atoms in a model without changing whether the model satisfies

the theory.

The authors define term symmetries to be those

symmetries that can be expressed as just a permutation of the

constants appearing in the atoms, rather than of the atoms

themselves.

They further define term-equivalent symmetries as

term symmetries that can be expressed as a partition of the

constants, with the symmetries being all permutations that map

each constant to another in the same block.

Such symmetries thus define an equivalence relation on models,

with equivalence being with regard to satisfiability, weight,

or another objective.

The idea of symmetry-breaking is to

introduce constraints to the theory that eliminate some

otherwise valid models, while making sure that at least one

model in each equivalence class survives.

If implemented as a

pre-processing step to an inference or satisfiability

algorithm, this would serve to reduce the search space.

QUALITY

At a first glance, the work appears to be of fairly high

quality.

The definitions are clear, and necessary theorems

are carefully stated and proved.

CLARITY

Having some knowledge of the field but not a deep background,

I found the exposition rather hard to follow.

While the basic

definitions and concepts are clear enough (if rather dense), a

running example domain would have been helpful to illustrate

the types of symmetries being considered.

Some of the results seem counter-intuitive, but this is not

discussed.

For example, the class of term symmetries being

larger than term-equivalent, one would expect that breaking

the former would result in a bigger speed-up than just

breaking the latter.

The opposite seems to be the case in

many experiments; the "Term" procedure sometimes takes longer.

Some clarification would be appreciated on whether this is

because the term symmetry-breaking propositions cause a

slow-down.

ORIGINALITY AND SIGNIFICANCE

While the paper presents original algorithms for detecting and

breaking term and term-equivalent symmetries, the significance

of these symmetry classes was not immediately clear.

An

argument should be made that these classes are sufficiently

rich and expressive to simplify real-world problems.

MINOR CORRECTIONS

In lines 188 and 189, "uniquely color" should be "unique

color". In line 88, "unique type" should perhaps be "single

type", since multiple constants can have the same type.

Summary: The work introduces techniques to detect and break

certain kinds of symmetries in statistical relational inference;

in particular the newly-defined classes of term and

term-equivalent symmetries.

While the algorithms appear sound

and efficient, the significance of these classes of symmetries

and the impact of breaking them is not completely

clear.

Submitted by Assigned_Reviewer_2

The paper attempts to solve an important problem that has not yet received enough attention (lifted MPE with evidence). The approach is very promising, by detecting and breaking symmetries at the first-order level.

The formalization for symmetry-breaking constraints coming from renaming permutations is an entirely novel contribution, and quite significant. I'm not sure how much this solving-by-grounding approach to lifted inference is promising in the long run, but it's an avenue worth exploring in much more detail.

The notion of term symmetries is not as novel as may be suggested by the paper. Identical or similar notions in the literature are called

* symmetry at the level of constants [Niepert, Lifted Probabilistic Inference: An MCMC Perspective, 2012]

* renaming permutations [Bui et al., Automorphism Groups of Graphical Models and Lifted Variational Inference, 2012]

* renaming automorphisms [Niepert, Symmetry-Aware Marginal Density Estimation, 2013]

* permutations of constants [Van den Broeck, PhD Thesis, 2013] While there may be some minor differences in how these ideas are formalized, the core ideas are the same, and this literature is too easily dismissed. I do appreciate the discussion of evidence in Section 3; it's just that the evidence problem in lifted inference is less related to the current paper than the references above.

Similar observations can be made about the notion of term equivalent symmetries and the algorithm for detecting these. Again, there is some similarity with ideas from the literature (e.g., lifted network construction, using saucy or nauty for finding relational symmeties).

I'm confused by lines 212-213: if you have to construct a node for each argument list that appears in evidence, then there are O(D^k) such nodes. Therefore the cost of finding term symmetries should be higher than what is stated on these lines, and identical to the direct approach.

The notation in Definition 4.4 is bad: using both C_i and C^i.

In the experiments (line 368), it's suspicious to report the best runtime over two MAXSat solvers.

Summary: This paper approaches lifted MAP inference from a new promising angle and cleanly formalizes symmetry-breaking constraints for relational (weighted) theories. This is sufficient to make the paper acceptable. On the other hand, the discussion of related work is not entirely satafying, especially about other techniques to find symmetries of constants: the most strongly related work by Niepert is not cited.

Submitted by Assigned_Reviewer_3

Symmetry breaking constraints are added to eg an MLN MAP instance. The instance is then ground and solved using an exact weighted partial MaxSAT algorithm such as Sat4j.

The paper is clearly written and contains enough information to allow one to replicate the techniques. The empirical results confirm that using symmetry breaking improves considerably over a 'vanilla' approach. Both "term-" and "term-equivalent" symmetry breaking is considered, providing similar performance.

However, it is difficult to be convinced that this method solved the sort of problems people care about. Also the empirical results are not that great. One typically cares more about being able to solve larger instances (eventually) than solving smaller instances a little more quickly. We need to see how much better the exact MAP solution was than, eg that produced by RockIt. Also each result for Tequiv and Term is the better of two times using MinMAXSAT and Sat4j, so each is not the result of running some particular algorithm unlike the case for RockIT.

The paper makes it sound as if symmetry breaking is a technique *only* used in SAT. Of course it is in fact used all over the place, particularly in constraint programming.

One issue that comes up in the literature is which of the following two options is superior: adding in symmetry breaking constraints vs pruning branches in the search using symmetry. This is not addressed here.

typos...

line 103 : simple sets -> simply sets line 136 : the method is exactly -> the method is exact line 189 : A uniquely color -> A unique colour line 244 : add missing "}" line 227 : The set all -> The set of all line 250 : above procedure -> the above procedure

After reading the authors' response:

Some of my concerns (and that of other reviewers) have been addressed and I have nudged up my overall rating for this paper accordingly.
Summary: A competent approach to symmetry-breaking is presented with modest empirical performance.

Author Feedback
Author rebuttal: Reviewer 1:
Term and term equivalent symmetries are prevalent in real world examples, eg entity resolution, social networks, comp bio etc (Gogate & Domingos 2011, Broeck et al 2011, Taghipour et al 2012, Bui et al. 2013).

We tested on the pigeon-hole problem because it exhibits term equivalent symmetries and a structure that often occurs in the very hardest reasoning problems, wherever there is a set of resources and a potentially greater set of demands (e.g. planning, scheduling, graph coloring, etc.). Pigeon-hole structure has also been shown to occur with high probability in very hard random constraint satisfaction problems (Chvatal and Szemeredi, 1988).

Term equivalent symmetry breaking can be more efficient than either term symmetry breaking or propositional symmetry breaking because a single term equivalent symmetry represents an exponential number of term symmetries (all permutations of term). A single term equivalent symmetry breaking formula, which is at worse linear in the number of ground atoms, is complete - it breaks an exponential number of symmetries. We will add details and an example.

Reviewer 2:
We agree that the term symmetries (permutations) have been introduced earlier in the literature. In fact, we have stated the connection with Bui et al.'s 2013 work: "Bui notes that symmetry groups can be defined on the basis of unobserved constants in the domain..". We have also cited the relevant papers by Broeck ([22],[23],[24]), and will add a reference to his thesis.

We are aware of the existing work by Niepert (2012, 2013) on permutation groups. We inadvertently left it out in our final submission due to some last minute editing. We will make sure the work is prominently cited.

Nevertheless, most existing work has not explicitly looked at the problem of finding term symmetries based on evidence. In the few cases that it has been considered, either only unary evidence has been dealt with efficiently (Broeck and Davis, 2012), evidence symmetries have been exploited in an approximate manner (Broeck 2013, Venugopal and Gogate 2014) or it has been reduced to an NP-hard problem (Hypercubes in Taghipour et al., 2012, Singla et al., 2014). We show that only looking at evidence, one can uncover a large class of symmetries and we give an efficient polynomial time algorithm for the same. This is a significant contribution.

Existing approaches using Saucy or Nauty find symmetries over the entire set of ground atoms. In contrast, we find symmetries over terms which can be a much smaller set (number of ground atoms is
exponential in the predicate arity). Regarding the issue on lines 212-213, we create a node only for those term combinations which appear explicitly in the evidence, which is typically much smaller
than all O(D^K) combinations.

As noted in the caption to Table 1, we tried Alchemy 2's lifted PTP algorithm, but it timed out on every instance. We also present results on MaxWalkSat and RockIt, but approximate methods, which one would expect to be faster than our exact method. We used sat4j for the pigeon hole and MiniMaxSAT for the advisor domain. SAT solvers use varying kinds of heuristics and different ones may be suited depending on the structure of the SBPs generated. For each domain, we used the one performing best since our goal was to show the power of SBPs. Exploring this further is a direction for future work.

We will fix inconsistent notation.

Reviewer 3:
Exploiting symmetries is important for a large class of application domains (please see reply to reviewer 1). It's true that we are ultimately interested in problems of much larger sizes. Nonetheless, a lot of such research is preceded by advances in exact solvers (e.g., belief propagation algorithm over trees giving rise to loopy BP, exact DPLL solvers giving rise to approximate solvers). Our current work focuses on symmetry breaking for exact MPE by exploiting the relational structure. Our experiments show the scalability of our approach compared to the vanilla algorithm which times out even for very small instances.

Table 1 shows that the running times for our exact method is comparable with that of the approximate solvers MaxWalkSat (MWS) and RockIt. Furthermore, the solutions we found were dramatically better: e.g. the violation scores of the solutions 1(a):
n us MWS
5 16 24
10 81 12,553
20 361 568,828
30 841 4,639,190
40 1,521 21,929,000
We will add this important data to the table.

We agree that symmetry breaking is not limited to SAT. Last paragraph of our related work section has some relevant discussion (see [26]). We will include more details in the camera ready. Whether to add SBPs or prune the search space is an important question. Ours is the first work exploring the former for relational theories and experimenting with pruning strategies is a part of future work.

Regarding the SAT solver issue, please see Reviewer 2. We will fix the typos in the final version.